# Colloidal and Acid Gelling Properties of Mixed Milk and Pea Protein Suspensions

**DOI:** 10.3390/foods11101383

**Published:** 2022-05-11

**Authors:** Isabelle Carolina Oliveira, Iuri Emmanuel de Paula Ferreira, Federico Casanova, Angelo Luiz Fazani Cavallieri, Luis Gustavo Lima Nascimento, Antônio Fernandes de Carvalho, Naaman Francisco Nogueira Silva

**Affiliations:** 1Center of Natural Sciences, Federal University of São Carlos (UFSCar), Buri 18290-000, SP, Brazil; isabelle.4862@gmail.com (I.C.O.); ferreira.iep@gmail.com (I.E.d.P.F.); angelo.lf.cavallieri@gmail.com (A.L.F.C.); 2Food Production Engineering Group, DTU Food, Technical University of Denmark, Søltofts Plads 227, DK-2800 Lyngby, Denmark; 3Department of Food Technology, Federal University of Viçosa (UFV), Viçosa 36570-900, MG, Brazil; luis.g.nascimento@ufv.br (L.G.L.N.); antoniofernandes@ufv.br (A.F.d.C.)

**Keywords:** dairy proteins, caseins, plant proteins, protein beverages, acid gel, colloidal stability

## Abstract

The present study aims to describe colloidal and acid gelling properties of mixed suspensions of pea and milk proteins. Mixed protein suspensions were prepared by adding pea protein isolate to rehydrated skimmed milk (3% *w*/*w* protein) to generate four mixed samples at 5, 7, 9, and 11% *w*/*w* total protein. Skimmed milk powder was also used to prepare four pure milk samples at the same protein concentrations. The samples were analyzed in regard to their pH, viscosity, color, percentage of sedimentable material, heat and ethanol stabilities, and acid gelling properties. Mixed suspensions were darker and presented higher pH, viscosity, and percentage of sedimentable material than milk samples. Heat and ethanol stabilities were similar for both systems and were reduced as a function of total protein concentration. Small oscillation rheology and induced syneresis data showed that the presence of pea proteins accelerated acid gel formation but weakened the final structure of the gels. In this context, the results found in the present work contributed to a better understanding of mixed dairy/plant protein functionalities and the development of new food products.

## 1. Introduction

According to the World Resources Report [1], the worldwide consumption of animal-source foods may increase 68% by 2050 due to the population growth. For this reason, the mixture of plant and animal proteins has been seen as an answer to the global demand for dairy and meat products in the upcoming decades. Field pea (*Pisum sativum*, L) is one of the plants that can be used as an alternative protein source. In general, pea seeds contain 10–20% (*w*/*w*) fiber, 40–50% (*w*/*w*) starch, and 18–30% (*w*/*w*) protein, which can vary according to environmental and genotypic factors. When compared to soy or other plant proteins, pea protein presents high digestibility, less allergenicity, low production cost, and high nutritional value [2,3]. However, when used in food systems, the poor water solubility and the pronounced taste, described as “beany”, limits its application [4,5]. Therefore, the implementation of pea protein into a mixed system with an animal protein source, such as milk, has the potential to produce nutritious foods with acceptable sensorial characteristics and distinctive functional properties.

In this sense, milk and dairy products are consumed by billions of people due to their industrial versatility, high nutritional value, and good taste. When used as an ingredient, milk is a major component of diverse dairy products, such as yogurt, butter, cheese, creams, and high-protein beverages. This diversity of application is possible because of milk proteins (mainly caseins and whey proteins), which are responsible for its functional properties. These include the ability to form gels, emulsions, and foams [6,7,8]. Mixed systems of milk and different plant proteins have been the focus of recent studies to evaluate their gelling and emulsifying properties [9,10,11,12]. The results demonstrate the possibility of creating innovative food products that can fulfill an emerging market of consumers who worry increasingly about sustainability and want to partially replace animal-based proteins [13].

Although the literature regarding the functional properties of food matrices containing plant and dairy proteins is growing [9,11,12,14], there is a disparity in the research between academic purposes and evaluations for industrial application. In this context, the present work aims to describe colloidal and acid gelling properties of mixed milk/pea protein suspensions, with total protein concentrations ranging from 3.0% (*w*/*w*) to 11.0% (*w*/*w*). Here, the main strategy was to carry out the experiments in a manner that can be replicated by the dairy industry. Regarding colloidal aspects, the suspensions were analyzed in relation to pH, viscosity, color attributes, amount of sedimentable material, and heat and ethanol stabilities. Concurrently, the acid gelling properties were evaluated using small oscillation rheology and induced syneresis.

## 2. Materials and Methods

### 2.1. Materials

Pea protein isolate (PPI) (NUTRALYS S85F) was provided by Roquette (Lestrem, France), and low heat skimmed milk powder (SMP) was provided by courtesy of Itambé (Sete Lagoas, Brazil). The PPI was composed of 83% *w*/*w* proteins, 3.8% *w*/*w* minerals and 0.1% *w*/*w* moisture; the composition of SMP was 35% *w*/*w* proteins, 50% carbohydrates, 7.5% minerals, and 4% *w*/*w* moisture. This information was provided by the manufacturers. Sodium azide was purchased from Sigma Aldrich (São Paulo, Brazil).

### 2.2. Methods

#### 2.2.1. Preparation of Suspensions

The SMP was rehydrated in distillated water with 3% *w*/*w* proteins. Sodium azide was added at 0.03% *w*/*w*, to prevent microbial growth, and the sample was stirred at room temperature for 1 h. To generate the mixed protein suspensions, the PPI was added to the rehydrated skimmed milk with 3% (*w*/*w*) proteins to achieve total protein concentrations of 5, 7, 9, and 11% (*w*/*w*). The same procedure was applied to prepare pure skimmed milk samples with the same total protein concentrations. In total, nine samples were produced, and the skimmed milk with 3% (*w*/*w*) proteins was used as the control (Table 1). All the protein suspensions were stirred overnight at room temperature and then stored at 4 °C prior to analysis.

#### 2.2.2. Colloidal Properties

##### pH of the Suspensions

The pH of the samples was measured at room temperature using a calibrated pH-meter, PHS-3E (Satra, São Paulo, Brazil).

##### Viscosity

Viscosity analyses were carried out using a controlled stress rheometer, MCR 102 (Anton Paar, Ostfildern, Germany). The rheological properties of protein suspensions were obtained under steady state shear at 20 °C with a stainless-steel cone-plane geometry (diameter 50 mm and cone angle 1°). The flow curves were determined using a shear rate within the range of 0 and 300 s^−1^. Three shear stress sweeps (up–down–up steps) were performed to verify the presence of shear time effects (thixotropy) [15,16]. Data from the third flow curve (steady state conditions) were adjusted to classical rheological models; the Newtonian Equation (1) or power law model Equation (2) was used according to the best coefficient of determination (R^2^).
σ = η.γ(1)
σ = k.γ^n^(2)
where σ is the shear stress (Pa), γ is the shear rate (s^−1^), η is the Newtonian viscosity, k is the consistency index (Pa.s^n^), and *n* (dimensionless) is the behavior index. The test was carried out in duplicate. For Newtonian fluids, the viscosity value was considered the Newtonian viscosity, while for shear thinning fluids, apparent viscosity was obtained at 60 s^−1^ of the third sweep.

##### Color

The color was measured using a Colorium7 Chromameter (Delta Color, São Leopoldo, Brazil) [17]. The equipment was calibrated with a blacklight trap and white calibration ceramic. The parameters, L* (luminosity), a* (redness/greenness), and b* (yellowness/blueness) were obtained through i7 software (Delta Color, São Leopoldo, Brazil). The test was conducted in duplicate.

##### Sedimentable Material (SM)

The amount of SM was determined by centrifugation (Centrifuge NT 820—Novatecnica, Piracicaba, Brazil) of 10 g of each sample in separate centrifuge tubes (15 mL), at room temperature for 30 min at 4856× *g*. To express the amount of SM as a percentage, the mass of the obtained pellets was divided by the mass of the sample submitted to centrifugation, and the result was multiplied by 100. The test was performed in duplicate.

##### Heat Stability

Thermal stability was determined by adding 2 mL of the protein suspensions in hermetically sealed glass tubes, followed by their immersion in an oil bath at 140 °C. The thermal stability was defined as the time (in seconds) until visual observation of protein coagulation commencement [18,19]. The test was made in triplicate.

##### Ethanol Stability

Ethanol stability was determined by mixing equal volumes of protein suspensions with ethanol solutions (Labsynth—São Paulo, Brazil) in Petri dishes with increasing ethanol concentrations. Ethanol concentrations ranged from 60 to 100% *v*/*v* and ascended at 2.5% intervals. The highest ethanol concentration without visual signs of coagulation was defined as the ethanol stability of the samples [18,19]. The test was carried out in triplicate.

#### 2.2.3. Gelling Properties

##### Gel Formation and Rheological Analyses

Before acidification, all milk samples and mixed protein suspensions were heated at 85 °C for 5 min and subsequently cooled to 42 °C. This step was intended to induce protein denaturation and consequent aggregation, which is a classic strategy employed by the dairy industry to improve the firmness of fermented milks. Then, gel formation was monitored in triplicate under isothermal conditions (42 °C) by oscillatory measurements in the same controlled stress rheometer equipped with the same cone-plate geometry used for steady state flow measurements (Section 2.2.2 Viscosity). Glucono delta-lactone (GDL) (Sigma Aldrich—São Paulo, Brazil) was added to the protein suspensions at 2% (*w*/*w*), followed by agitation for 1 min, and then transferred to the rheometer. Time sweeps were conducted at an oscillation frequency of 0.1 Hz within the linear viscoelastic domain (0.1% of oscillatory strain). The gel point (tg) was considered as the time when the crossover between elastic (G’) and viscous (G”) moduli took place, in accordance with Cavallieri and da Cunha [20].

##### pH Kinetics

During the acidification of the samples, the pH was measured at 42 °C after 20 min intervals using a calibrated pH meter PHS-3E (Satra, Brazil).

##### Induced Syneresis

GDL was added to 15 g of protein suspensions at 2% (*w*/*w*) and at 42 °C in cylindrical plastic flasks (−1.2 cm in diameter), followed by agitation for 1 min. The flasks were kept under rest at 42 °C for 4 h. Immediately after this gelling process, the induced syneresis was determined as the percentage of the liquid drained after centrifugation (NT820—Novatecnica, Brazil) of the gels at 1800× *g* for 30 min [21].

#### 2.2.4. Statistical Analysis

The experiments were replicated three independent times. In each run, the batch of skimmed milk at 3% (*w*/*w*) proteins was split into nine samples and treatments were assigned to the samples at random. Firstly, the data were explored through descriptive statistics (means, standard deviations, and plots). Then, second-degree polynomials were fitted to describe the effect of protein concentration on the colloidal characteristics of the samples. Non-significant terms were removed from the model during a stepwise procedure, and the best model was selected according to the lowest AIC (Akaike’s Information Criterion). For each fitted model, the significance was assessed using Wald’s test, the explanatory power was given by the adjusted coefficient of determination, and the F-test was carried out to evaluate the lack of fit. The adequacy of the fitted models was examined through residual analyses, and the normality and homogeneity of variances were evaluated by Shapiro–Wilk’s and Levene’s tests, respectively. The Box-Cox’s optimal transformation was applied to the values of viscosity to normalize the data and stabilize the variance. The fittings were plotted with the original data and error bars (SE, *n* = 3 per treatment). The fitted models, the adjusted coefficients of determination, the regression mean-squared errors, and F-tests for the lack of fit are provided as in Appendix A.

There was no suitable polynomial model for describing the relation between heat stability and protein concentrations for milk samples and mixed protein suspensions. Thus, the heat stabilities of these samples were compared within each protein dosage by a Student’s *t*-test. A power analysis was conducted considering similar scenarios to those verified in our data; in all the simulations, the *t*-test ensured 85% of power or more on differentiating the means of experimental groups.

The statistical hypotheses were tested considering a significance level of 5%. All the statistical analyses were conducted using R software [22]. The research data are available at an open access repository [23].

## 3. Results and Discussion

### 3.1. Colloidal Characterization of the Suspensions

#### 3.1.1. pH

The pH data are shown in Figure 1. The pH of milk samples decreased as the total protein concentration increased. Compared to mixed protein suspensions, the lower pH values found for milk samples were due to the presence of acidic compounds naturally found in milk, such as caseins, whey proteins, and phosphate salts [6]. In contrast to the samples containing only milk proteins, the mixed suspensions exhibited stable pH values despite the increase in protein concentration. This has not been commonly reported in other studies evaluating pH levels of pea protein suspensions as a function of protein concentration. In general, this is explained by the fact that the pH values of isolated pea protein suspensions are corrected before performing further experiments. However, Jiang et al. [24] used the same PPI applied in the present work (NUTRALYS S85F) and described the pH as being close to 7.0 of PPI suspensions at 25–30 g protein per liter, which is consistent with the results presented in Figure 1.

#### 3.1.2. Viscosity

Figure 2 highlights the viscosity values of the suspensions as a function of the total protein concentration. Milk samples and mixed protein suspensions exhibited Newtonian behavior up to 9% (*w*/*w*) protein (Appendix A); therefore, the Newtonian viscosity of these samples was used in Figure 2. For the samples at 11% (*w*/*w*) protein, the apparent viscosity presented in Figure 2 was obtained at a shear rate of 60.0 s^−1^. In fact, a pseudoplastic behavior was observed for these two highly concentrated samples (Appendix A), with a behavior index of 0.86 ± 0.02 for mixed protein suspensions and 0.87 ± 0.06 for milk. The consistency index was 0.11 ± 0.02 Pa.s^n^ for the mixture and 0.05 ± 0.03 Pa.s^n^ for milk. Similar results were observed for skimmed milk [25] and equal mixtures of pea and milk protein [26] at similar protein concentrations.

As shown in Figure 2, the viscosity increased with total protein concentrations for both systems. Nevertheless, the increase in viscosity was higher in the presence of pea protein levels above 7% (*w*/*w*). The viscosity results for skimmed milk samples are in accordance with the literature, as the viscosity of milk increases as a function of the total solids content [27,28]. For the mixed protein suspensions, it is plausible that the low solubility of pea proteins [29] was responsible for the higher viscosity values. Indeed, the presence of aggregates or insoluble material increases the volume fraction of protein suspensions, which consequently increases the apparent viscosity for a given temperature and the shear rate [30]. This result is in line with the determination of the amount of sedimentable material (Section 3.1.4).

#### 3.1.3. Color

Figure 3 shows the parameters L*, a*, and b* of the samples. An increase in the lightness (L*—Figure 3A) of the samples containing only milk proteins (Figure 3A) as a function of the protein concentration was observed. This increase in L* is due to the increment of particles scattering light, mainly casein micelles [31]. The parameter a* (Figure 3B) is correlated with the presence of absorbing compounds that reflects red (positive values) and green (negative values) light. In milk samples, the main molecules responsible for parameter a* are lactoferrin for redness and riboflavin for greenness [27]. In general, the values of a* are negative for milk and dairy protein suspensions, regardless of the protein/fat concentrations and the physical treatments applied to milk [27,28]; which is consistent with the results found in the present work. The parameter b* (Figure 3C) is associated with the presence or absence of molecules that reflect yellow or blue. In the case of milk, the carotenoids associated with the fat portion are responsible for its yellow tendency [27]. As a matter of fact, all industrially produced skimmed milk using centrifuges present a residual fat portion. Therefore, the increase in the b* parameter found in this work for skimmed milk can be attributed to an increase in the fat concentration of the samples.

The L* values for mixed suspensions decreased as a function of the protein concentration, which can be attributed to the increase in the concentration of dark pigments, e.g., phenolic compounds naturally present in pulses [32]. The increase in pea protein concentration in the mixed suspensions also caused the augmentation of a* and b* parameters (Figure 3A,B). According to Saldanha do Carmo et al. [33], independent of the dehulling method applied to the production of protein-rich fractions from pea beans, the color compounds are naturally present in the resulting powder and are related to pea variety used to produce the PPI. These data are important since color is a sensorial criterion that influences consumer’s preferences. Thus, when developing dairy products with pea proteins, technologists and/or engineers must take this aspect into account because the products will present a darker color than usually observed for dairy products.

#### 3.1.4. Sedimentable Material (SM)

The percentages for SM are shown in Figure 4. The results show that the SM percentages were quite stable for all milk samples. However, the SM percentages were linearly correlated with the total protein content of the mixed protein suspensions and were significantly higher than those of the milk samples. This result was somewhat expected since only the albumin fraction of the PPI is water soluble, and it accounted for approximately 20% of the total protein content of commercial PPI [2]. In fact, some research groups work only with the soluble fraction of pea proteins when searching for new functionalities in association with dairy proteins, which can be considered as a valid scientific approach [12,34]. In the present study, however, we decided to use PPI powder as a whole because this strategy is closer to an industrial application of PPI in mixed protein suspensions. The separation of the soluble fraction from PPI, or pea flour, is a process that requires time, resources, and chemicals [32]. In addition, the disposal of the insoluble protein fraction should also be considered. On one hand, the insoluble protein fraction can be undesirable if the industrial goal is to produce protein beverages. On the other, this insoluble fraction is not necessarily a problem if the food product is more solid-like (elastic), such as some types of fermented milk products, such as cheese analogues, ice cream, and desserts.

#### 3.1.5. Heat Stability

The results for heat stability are presented in Table 2. For both systems, the heat stability decreased insofar as the total protein content increased. The results for the milk samples are comparable to those reported by Dumpler and Kulozik [35] and Huppertz et al. [19] for similar pH values, protein concentrations, and temperatures. In effect, caseins are considered rheomorphic proteins that are naturally resistant to heating, without presenting a well-defined tertiary structure [36]. Moreover, caseins are present in milk as supramolecular aggregates, known as casein micelles, which are electrostatically stabilized by an external layer of κ-casein that prevents their aggregation. At high temperatures (above 140 °C) for extended periods (several minutes), milk coagulates as a result of the collapse of the external layer caused by a reduction in pH originating from lactose degradation [37]. As can be inferred from Figure 1 and Table 2, the milk sample at 11% (*w*/*w*) protein is less heat stable and more acidic (lower pH) than the milk sample at 3% (*w*/*w*) protein. Nevertheless, pH reduction is not the only factor contributing to decreased heat stability. It is also necessary to consider that the ionic calcium concentration of the milk samples naturally increased as a function of the total protein concentration. This was due to SMP being employed to prepare these samples. Indeed, the increase in ionic calcium concentration induced a reduction of casein micelle stability [38] and contributed to the reduction of heat stability of the milk samples (Table 2).

The mixed protein suspensions presented significantly lower heat stability than the skimmed milk samples (*p* < 0.05). Pea proteins are mainly composed of globulin and albumin fractions, which exhibit well-defined secondary and tertiary structures that are naturally more sensitive to heating [2,36]. High temperatures induce the denaturation of pea proteins, which leads to hydrophobic and covalent inter-protein interactions, resulting in aggregation and precipitation [32,39]. However, there are no comparative results in the literature, since this is the first work evaluating heat stability of mixed protein suspensions. It is noteworthy that if the mixed suspensions were acidified (pH ~6.0 or lower), they would tend to form heat-induced gels during heating instead of precipitating, as observed by Schmitt et al. [40].

#### 3.1.6. Ethanol Stability

The data for ethanol stability are shown in Figure 5. The ethanol stability test is globally employed by the dairy industry to indirectly evaluate the heat stability of milk prior to pasteurization or commercial sterilization. In fact, as an organic solvent, ethanol reduces the dielectric constant of the aqueous medium, which in turn reduces the hydration of the external layer of casein micelles and diminishes their stability [31]. This effect is more pronounced when the pH is low and the concentration of ionic calcium is high [36,37]. For the milk sample at 3% (*w*/*w*) protein, ethanol stability was 92.5%, in accordance with Gulati et al. [41]. For other milk samples, the ethanol stability linearly decreased as a function of total protein concentration, as was observed for heat stability (Section 3.1.5).

In the case of the mixed protein suspensions, they exhibited higher ethanol stability than milk samples at intermediate protein concentrations (from 5 to 9% *w*/*w*). The increase in ethanol stability caused by the addition of pea protein might be due to the capacity for pea proteins to bind calcium. Silva et al. [12] showed that the concentration of ionic calcium diminished from 6 mM in casein micelles suspensions to approximately 2 mM when 6% (*w*/*w*) pea protein was added at pH 5.8. The authors concluded that the pea proteins acted as chelating agent for calcium ions. However, at 11% *w*/*w* protein, the stability of the mixed protein suspension was lower than that of milk samples at the same protein concentration. In effect, another factor that might have contributed to the reduction in ethanol stability for the more concentrated milk samples and mixed protein suspensions was the high concentration per se. It is well established that the increase in the concentration of colloidal particles enhances the probability of collisions due to Brownian motion. Therefore, the particles in concentrated protein suspensions, under destabilizing conditions, such as high temperature or high ethanol concentration, are more likely to interact with each other, increasing the aggregation and precipitation rates [31].

### 3.2. Rheological Analysis of the Gels

The experimental conditions for studying the acid gelling properties of milk and mixed suspensions were set in order to simulate industrial yogurt production taking place at 42 °C for 4 h. Therefore, Table 3 presents the overall characteristics of the gels, including the gel point (tg), the complex shear modulus (G*) at tg, the final G* and pH (obtained after 240 min of acidification), and induced syneresis. Figure 6 presents the evolution of G* against the reduced time (t/tg). Figure 7 assembles the evolution of G* and pH as a function of time after GDL addition for 240 min.

#### 3.2.1. Initial Gelation Stage

As shown in Table 3, the start of gelation (tg) for milk samples was delayed as protein content increased. Although the pH was not measured at tg, the increase in buffering capacity of the more concentrated milk samples can explain the longer periods required to achieve tg. In the present study, SMP was used to increase the protein content of milk samples. Thus, in addition to the buffering capacity of caseins and whey proteins, there is a significant contribution of milk salts, such as colloidal calcium phosphate and soluble phosphate, carbonate, and citrate [42]. Concomitantly, the values for G* at tg slightly increased in milk samples, which can be related to the higher amount of protein per unit volume, allowing for more protein–protein interactions at the beginning of the gelation process. When comparing milk and mixed suspensions, it was observed that the presence of pea proteins reduced tg. Similar results were noticed during the GDL-induced acidification of 14.8% *w*/*w* mixed (1:1 ratio) milk and pea protein suspensions at 30 °C for 20 h in the presence of 1.0% (*w*/*w*) NaCl [9]. The authors attributed the decrease in tg to a lower buffering capacity of suspensions containing pea proteins [9]. It is noteworthy that, here, the viscosity of mixed suspensions was higher than that of milk samples at comparable protein contents (Figure 2) and could partially contribute to the reduction in tg in the presence of pea proteins. Regarding the mixed suspension at 11% (*w*/*w*) protein (Table 3), the elastic modulus (G’) was higher than the viscous modulus (G”) during the initial moments of rheological analysis. Afterwards, it was not possible to determine tg; i.e., this sample was already gelled when the rheometer started to measure its rheological properties.

The variation in the complex shear modulus (G*) during gelation is shown against t/tg in Figure 6. Indeed, shifting the data along the horizontal axis eliminates the effect of the GDL hydrolysis kinetic, and the representation of G* against the reduced time (t/tg) makes it possible to differentiate the gels with respect to their structure [20]. Therefore, the results demonstrated that the structure formation of the gels containing pea proteins are distinctive of those formed by only milk proteins, and that the protein concentration had a significant effect on the gel structure (Figure 6). In general, for both systems, the increase in the protein concentration leads to more elastic gels. However, the rates at which the structure of pure milk gels evolve are much faster than those of the mixed samples (Figure 6) despite similar initial G* values at tg (Table 3). This kind of antagonistic effect between milk and plant proteins on the acid gelling properties was also reported by Ben-Harb et al. and Roesch et al. [9,43]. They suggested that the formation of different-sized protein aggregates during heating and acidification was correlated with the formation of large pores in mixed protein matrices, leading to more brittle acid gels. Moreover, according to Mession et al. [34], legumin aggregates formed during heating and acidification of pea proteins have a low potential for producing non-covalent interactions, which disturb the formation of more structured and elastic gels.

#### 3.2.2. Evolution of Gelling Properties

The viscoelastic properties of milk and mixed gels obtained under 0.1% of oscillatory strain and at 0.1 Hz after 240 min following GDL addition are shown in Figure 7. The results for acid milk gels (Figure 7A,C) were in accordance with the literature for experiments performed under similar protein concentrations, pH, and temperature [44,45,46]. It should be highlighted that, during the production of fermented milks, the heat treatment applied to milk prior to acidification induces the formation of whey protein/casein aggregates that increases the firmness of the acid gels compared to those obtained from non-thermally treated milks [44,45]. In the present work, the acid milk gels became more elastic as the protein content increased (Figure 7A and Table 3), which is a direct result of a denser protein matrix in association with the heating step performed before acidification.

Concerning the mixed gels, the viscoelastic behavior during acidification (Figure 7B,D) depended on the pea protein concentrations, which corroborates previous studies [12,43]. For the same total protein content, the mixed gels were less elastic than pure milk gels (Figure 7A,C; Table 3). Although a certain degree of co-gelation between whey and pea proteins in the mixed gels can be assumed [47], this result can be attributed to the fact that plant and milk proteins present a competitive dynamic, forming independent networks that do not strongly interact in acidic conditions [40]. Furthermore, increasing the proportion of plant protein in acid mixed gels leads to the formation of large pores in the protein matrix, reducing the strength of the gels [9,43]. Thus, the formation of large pores can be related to the reduction in G’ for mixed gels at 11% (*w*/*w*) protein (Figure 7B and Table 3).

Another important aspect of protein gels is pH. Here, the evolution of pH as a function of acidification time is represented in Figure 7E,F for milk and mixed gels, respectively; final pH values are given in Table 3. For both samples, the higher the protein concentration, the higher the final pH. However, for equal protein concentrations, the milk gels presented higher values than the mixed ones. This effect can be attributed to a greater buffering capacity of milk proteins and salts (as discussed in Section 3.2.1). Additionally, except for the milk sample at 3% *w*/*w* and the mixed sample at 5% *w*/*w* protein, the pH values of the other samples were close to the average isoelectric point of milk and pea proteins (~4.5–4.6) [6,48]. This information is relevant insofar as the differences in the gelling properties between milk and mixed samples observed in the present work did not have a major influence on pH.

Lastly, the induced syneresis results are presented in Table 3. Syneresis is an important sensorial aspect of fermented milks that is related to the rearrangement and shrinking of the protein matrix under acidic conditions. With respect to the milk gels, the total protein concentration greatly influenced syneresis, varying from 72.1% at 3% (*w*/*w*) protein to 1.5% at 11% (*w*/*w*) protein concentration. In this case, a highly hydrated and denser protein network, with a consequently low porosity, can explain the improvement in water holding capacity of the milk gels as a function of the total protein content [49]. Regarding the mixed gels, even though the protein concentration had a significant effect on induced syneresis, the results were more limited compared to the milk samples. Table 3 shows that the mixed gel at 5% (*w*/*w*) protein exhibited lower induced syneresis than milk gels at 3 and 5% (*w*/*w*) protein concentration. For mixed gels from 7 to 11% (*w*/*w*), the induced syneresis was higher than those of milk gels. As observed by Yousseef et al. [50], a partial replacement of milk by pea proteins can intensify the syneresis in fermented milks obtained from diverse lactic acid bacteria. Overall, syneresis is not necessarily correlated with gel strength [49]. Additionally, its reduction as a function of the total protein content for both types of gels in the present study can be attributed to the increase in structural (nonsolvent) and immobilized water molecules that are present in open pores of the gelled matrix and between chains of pea and milk proteins [31].

## 4. Conclusions

The experimental strategy adopted in the present work was established in a manner that can be replicated by the food industry to develop new products that combine high-quality plant and dairy proteins. The data analysis produced models that accurately describe the physicochemical behavior of milk and mixed protein suspensions, including food attributes that are important for industry. These include characteristics such as color, viscosity, and heat and ethanol stabilities. The equations that predict these characteristics are available in the Appendix A. In this regard, mixed suspensions were darker and presented higher pH, viscosity, and percentages of sedimentable material than milk samples. At the same time, heat and ethanol stabilities were similar for both systems and were reduced as a function of total protein concentration. The rheological analysis of the acid-induced gels made from milk and pea protein suspensions showed that the presence of pea proteins accelerates gel formation but weakens the structure of mixed gels compared to pure milk gels. Taken together, the results presented in this study can be useful for the current transition towards more plant-based foods and contributes to the comprehension of the functional properties of mixed dairy/plant protein systems, such as gel, emulsion, and foam formation. Notably, the insoluble fraction of PPI can be a sensorial problem to the development of a non-viscous mixed milk/pea protein beverage. Notwithstanding, for the production of gelled foods, this insoluble fraction can be integrated into the gel matrix without negative consequences to the sensorial acceptance of the final product.

## Figures and Tables

**Figure 1 foods-11-01383-f001:**
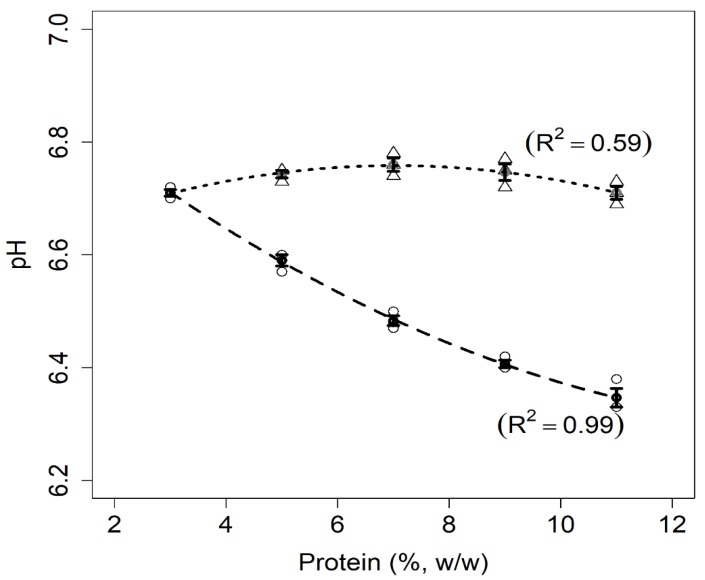
Relationship between pH and protein concentrations for milk (dashed line) and mixed protein (dotted line) samples. The figure shows the data points for milk (circles) and mixed samples (triangles), error bars (SE), fitted curves, and adjusted coefficients of determination (R^2^).

**Figure 2 foods-11-01383-f002:**
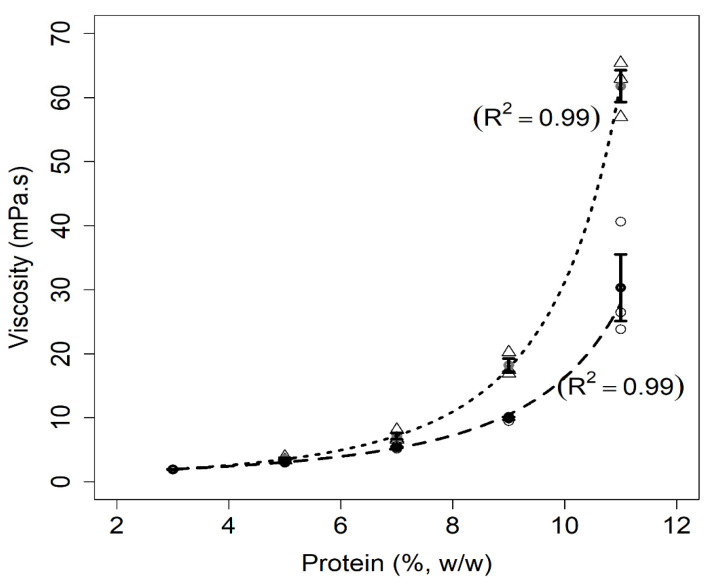
Relationship between viscosity and protein concentrations for milk (dashed line) and mixed protein (dotted line) samples. The figure shows the data points for milk (circles) and mixed samples (triangles), error bars (SE), fitted curves, and adjusted coefficients of determination (R^2^).

**Figure 3 foods-11-01383-f003:**
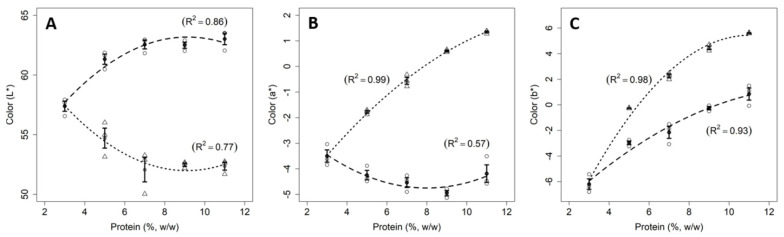
Relationship between color parameters and protein concentrations for skimmed milk (dashed line) and mixed protein (dotted line) samples. The figure shows the data points for milk (circles) and mixed samples (triangles), error bars (SE), fitted curves, and adjusted coefficients of determination (R^2^). The parameters L* represents lightness (**A**), a* red (+), and green (−) components (**B**), and b* yellow (+) and blue (−) components (**C**).

**Figure 4 foods-11-01383-f004:**
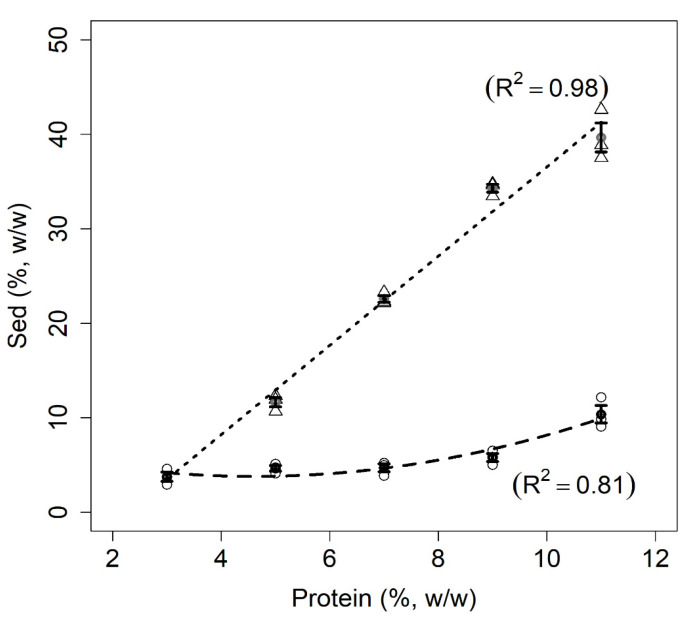
Relationship between sedimentable material (Sed % *w*/*w*) and protein concentrations for milk (dashed line) and mixed protein (dotted line) samples. The figure shows the data points for milk (circles) and mixed samples (triangles), error bars (SE), fitted curves, and adjusted coefficients of determination (R^2^).

**Figure 5 foods-11-01383-f005:**
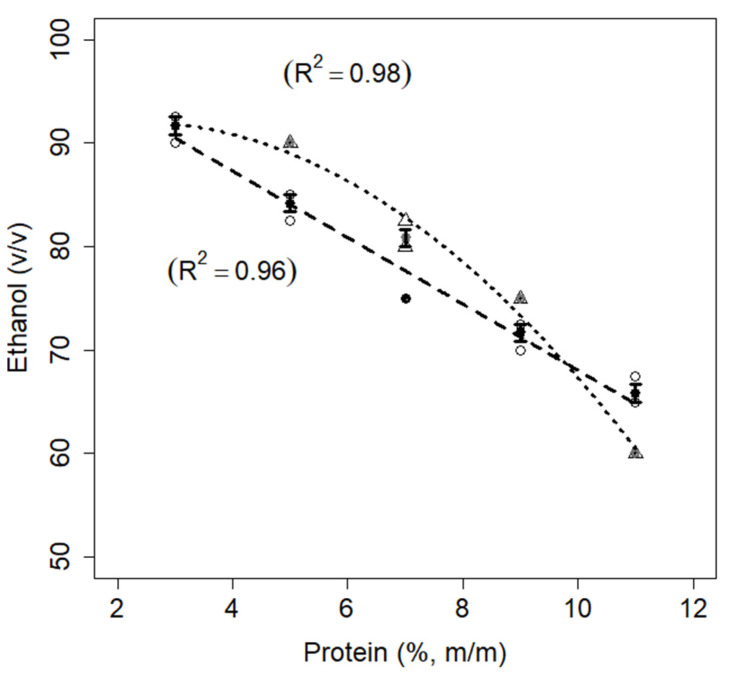
Relationship between ethanol stability and protein concentrations for skimmed milk (dashed line) and mixed protein (dotted line) samples. The figure shows the data points for milk (circles) and mixed samples (triangles), error bars (SE), fitted curves, and adjusted coefficients of determination (R^2^).

**Figure 6 foods-11-01383-f006:**
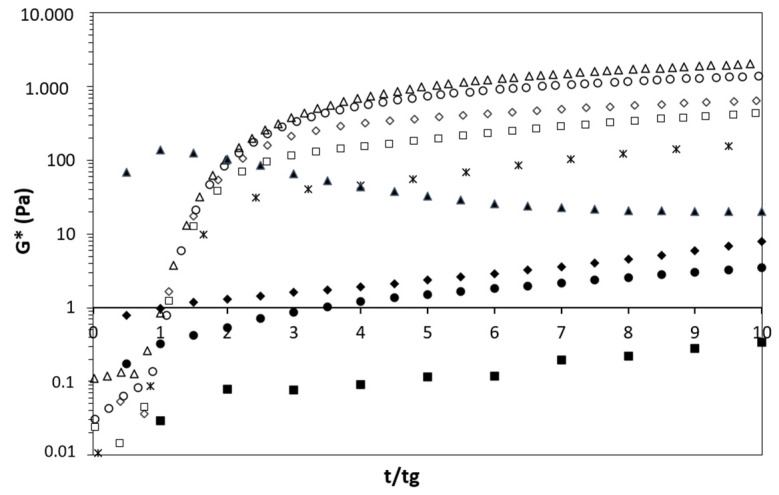
The evolution of the complex shear modulus (G*) as a function of reduced time (t/tg) for milk (open symbols) and mixed suspensions (closed symbols). The square, diamond, circle, and triangle markers represent the 5, 7, 9, and 11% (*w*/*w*) protein concentrations, respectively. The control milk sample is represented by the “x” symbol.

**Figure 7 foods-11-01383-f007:**
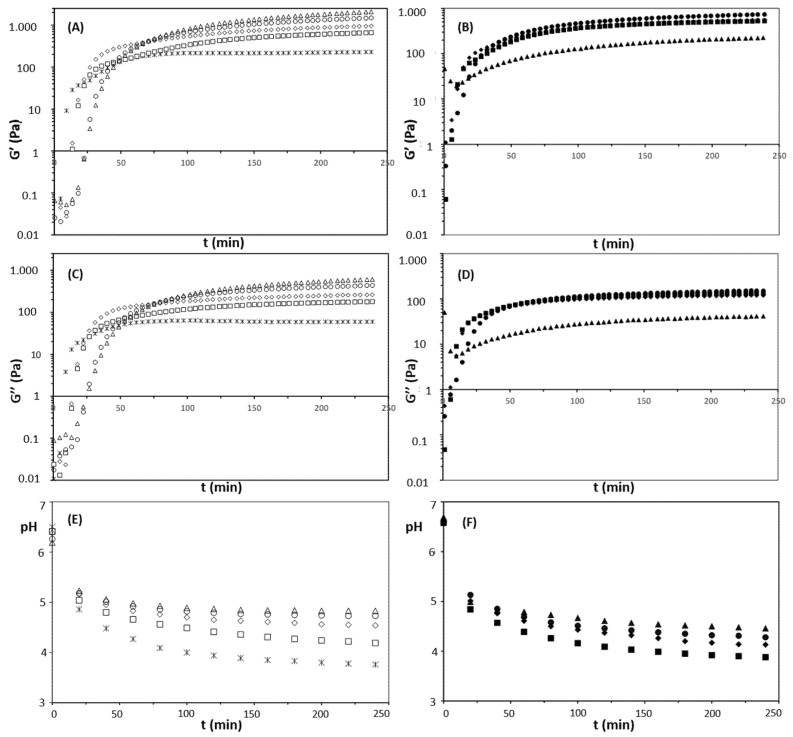
Evolution of elastic (G′) (**A**,**B**) and viscous shear moduli (G″) (**C**,**D**), and pH (**E**,**F**) as a function of acidification time (t) for milk (open symbols) and mixed suspensions (closed symbols). The square, diamond, circle, and triangle markers represent 5, 7, 9, and 11% (*w*/*w*) protein, respectively. The control milk sample is represented by the “x” symbol.

**Table 1 foods-11-01383-t001:** Protein composition of the samples used in the present work.

Samples	Milk Protein (%)	Pea Protein (%)	Total Protein (%)
Control	3	-	3
Skimmed milk	5	-	5
	7	-	7
	9	-	9
	11	-	11
Mixed protein suspensions	3	2	5
	3	4	7
	3	6	9
	3	8	11

**Table 2 foods-11-01383-t002:** Heat stability (in sec.) of milk samples and mixed protein suspensions represented as mean (±standard error), and comparisons between milk and mixed protein suspensions within each protein concentration according to a Student’s *t*-test.

Protein % (*w*/*w*)	Milk (sec.)	Mixed Suspensions (sec.)	*t*-Statistics	*p*-Values
3	1314 ± 61	--	--	
5	506 ± 27	380 ± 25	3.44	0.026 *
7	591 ± 5	284 ± 35	8.63	0.000 ***
9	414 ± 5	178 ± 5	35.48	0.000 ***
11	176 ± 6	144 ± 8	3.16	0.034 *

Signif. Codes: * *p* < 0.05; *** *p* < 0.001.

**Table 3 foods-11-01383-t003:** Gel point (*t_g_*), complex modulus (G*), and corresponding pH for mixed protein suspensions and milk samples.

Protein Concentration % (*w*/*w*)	*t_g_* (sec.)		G* at *tg* (Pa)		Final G* (Pa)		Final pH		Induced Syneresis % (*w*/*w*)	
	Milk	Mixed	Milk	Mixed	Milk	Mixed	Milk	Mixed	Milk	Mixed
3	240 ± 138	-	0.16 ± 0.18	-	227.6 ± 14.7	-	3.76 ± 0.04		72.1 ± 1.9	-
5	642 ± 84	90 ± 48	0.27 ± 0.07	0.11 ± 0.03	688.5 ± 24.3	535.9 ± 23.7	4.19 ± 0.05	3.88 ± 0.04	62.0 ± 1.5	49.6 ± 6.7
7	834 ± 216	48 ± 6	0.32 ± 0.14	0.90 ± 0.12	954.6 ± 162.4	545.3 ± 34.9	4.54 ± 0.03	4.13 ± 0.05	32.9 ± 8.1	46.5 ± 0.45
9	1152 ± 126	54 ± 12	0.45 ± 0.16	2.77 ± 3.77	1054.8 ± 160.1	900.7 ± 298.4	4.73 ± 0.01	4.28 ± 0.04	2.3 ± 0.2	46.5 ± 8.7
11	1140 ± 288	-	0.60 ± 0.35	-	2076.9 ± 144.2	278.0 ± 68.4	4.83 ± 0.01	4.46 ± 0.03	1.5 ± 0.5	25.9 ± 1.1

## Data Availability

The research data are available at an open access repository: https://data.mendeley.com/datasets/bw4sxsyg37/1, accessed on 9 May 2022.

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
