# Peer review of "Colloidal and Acid Gelling Properties of Mixed Milk and Pea Protein Suspensions"

_foods, 2022, doi:10.3390/foods11101383_

Round 1

Reviewer 1 Report

The authors investigated the gelling properties of mixed suspensions of pea proteins and milk performed under experimental conditions based on the environment of food industry. The results obtained in this work may contribute to the development of new food products.

Minor points:

  • “Acidification time (t/tg)” in the caption of Fig. 7 would be “acidification time (t),”
  • “The ability to form gels, emulsions e foams” (line 461) would be “the ability to form gels, emulsions and foams”

Author Response

The authors investigated the gelling properties of mixed suspensions of pea proteins and milk performed under experimental conditions based on the environment of food industry. The results obtained in this work may contribute to the development of new food products.

Minor points:

1. “Acidification time (t/tg)” in the caption of Fig. 7 would be “acidification time (t),”

It was corrected in the caption of Fig. 7 (line 462).

2. “The ability to form gels, emulsions e foams” (line 461) would be “the ability to form gels, emulsions and foams”

It was corrected in the new version (line 481).

Reviewer 2 Report

The manuscript by Oliveira et al. studied the gelation of mixed milk and pea protein suspensions. The following points should be addressed.

  1. It would be better to give typical curves of shear viscosity-shear rate. It is not clear why the viscosity of 11% sample is taken at 60s-1.
  2. What is the fitting function used in Fig. 2, polynomial? If it is not the exponential function, the statement “the viscosity increased exponentially” should be corrected.
  3. It would be more meaningful to use a well-known rheological model to fit the data in Fig. 2.
  4. What do the sedimentable materials contain? Only water-insoluble fraction of PPI?
  5. The accuracy in determining the gel point is critical to understanding the process. It is necessary to supply the evolution of G’ and G” at least for typical samples.
  6. Table 3 shows a significant decrease in the gelation time, an increase of modulus at the gelation point, and a decrease in the final modulus in the mixed system. Although some discussion is given, it is necessary to provide additional evidence. A strange point is that the fast gelation in 9% sample disappears in 11% sample.

Author Response

RESPONSE TO REVIEWER 2 COMMENTS

The manuscript by Oliveira et al. studied the gelation of mixed milk and pea protein suspensions. The following points should be addressed.

1. It would be better to give typical curves of shear viscosity-shear rate. It is not clear why the viscosity of 11% sample is taken at 60s-1.

In fact, the first part of the paper was focused on the relationship between colloidal properties and protein concentration. In this sense, to be coherent with the modelling applied to the other physicochemical characteristics (pH, color, sedimentable material, heat and stability) and to keep the conciseness of the paper, we decided to present the fitted models describing the effect of protein concentration on the viscosity values. However, we understand that flow curves are an important information and suggest to present the curves of viscosity-shear rate as supplementary material. Therefore, we included a supplementary material file (called “Supplementary material 2”) with one graphic showing the flow curves for Newtonian samples (from 3 to 9% protein) and another one for non-Newtonian samples (11% protein). The mention to Supplementary Material appears in lines 204 and 207/208.

Concerning the viscosity at 60 s-1 for 11% protein samples, we choose this shear rate because it is close to shear rate in the oral cavity during mastication. As the other samples presented a Newtonian behavior, the viscosity values were the same regardless the shear rate.

2. What is the fitting function used in Fig. 2, polynomial? If it is not the exponential function, the statement “the viscosity increased exponentially” should be corrected.

The reviewer is right, the fitting function is polynomial, and the word “exponentially” was removed (line 212).     

3. It would be more meaningful to use a well-known rheological model to fit the data in Fig. 2.

In this new version, these well-known rheological models for viscosity are presented in the Supplementary Material 02. For 11% protein samples, which presented a pseudoplastic behavior, we use a power-law model to describe the relationship between shear stress and shear rate. Regarding the viscosity of the Newtonian samples, it was fitted using linear equations. These models are described in the Material and Methods section, from line 92 to line 100.

4. What do the sedimentable materials contain? Only water-insoluble fraction of PPI?

In order to keep our work focused on industrial reality, we decided to use the PPI as a whole and we did not fractionate the soluble proteins. The goal of measuring a sedimentable fraction was to estimate the water-insoluble fraction of PPI because during the development of a mixed milk/pea protein beverage, this insoluble fraction can be a sensorial problem. However, concerning the development of a gelled food, this insoluble fraction could be integrated into the gel matrix, without negative consequences to the sensorial acceptance of the product. As we focused on the suspension as a whole, we did not perform further analyses on the sediment obtained by centrifugation.

5. The accuracy in determining the gel point is critical to understanding the process. It is necessary to supply the evolution of G’ and G” at least for typical samples.

As a matter of fact, we used only G* in our illustrations (Fig. 6 and 7) in order to have clearer graphics. Nevertheless, we also recognize that the presentation of G’ and G’’ is important for the readers. Thus, as G* values are already presented in Table 3 and Fig. 6, we replaced G* in Fig. 7 by G’ and G’’ (line 460).

6. Table 3 shows a significant decrease in the gelation time, an increase of modulus at the gelation point, and a decrease in the final modulus in the mixed system. Although some discussion is given, it is necessary to provide additional evidence. A strange point is that the fast gelation in 9% sample disappears in 11% sample.

We understand the Reviewer’s viewpoint and his/her concerns about tg for the mixed system at 11% protein. Indeed, we were not able to determine tg for this samples because when the rheometer started to measure G’ and G’’, the system was already gelled. As stated from line 137 to line 135, “GDL was added to the protein suspensions at 2 % (w/w), followed by agitation for 1 minute, and then transferred to the rheometer”. It is plausible that the crossover between G’ and G’’ took place during this period.

In this regard, we wrote the following sentence in the paper (lines 381 – 385): “Regarding the mixed suspension at 11% (w/w) protein (Table 3), the elastic modulus (G’) was higher than the viscous modulus (G”) since the initial moments of rheological analysis, and then it was not possible to determine tg, i.e., this sample was already gelled when the rheometer started to measure its rheological properties.”

Reviewer 3 Report

Dear Authors, 

The article penned by  Oliveira et al. is interesting and important to the industry. its strong point is the discussion of the results. However, I have a linguistic comments. in this form the article cannot be accepted. The text has to undergo English editing by native speaker in terms of grammar and  punctuation. The article in current form is not a reader friendly.  there are a lot of errors in the text, there is no point in listing them all, I will  just to name a few. Errors  if not corrected will detract from your interesting paper. There are many commas in the text that are used unjustifiably.  Also, please don't use this many semicolons, combine sentences with each other more neatly. I recommend major revision.

Some other comments to improve the manuscript:

Line 29 -  partial solution to the global demand

Line 36 -…as beany limit its application

Line 45 – there is a lack of current publications on special product fortification technologies with whey protein preparations and using their functional properties. please cite:

Shayanti Minj and Sanjeev Anand (2020) Whey Proteins and Its Derivatives: Bioactivity, Functionality, and Current Applications

Kristensen et al (2021) Improved food functional properties of pea protein isolate in blends and co-precipitates with whey protein isolate

Ghanimah (2018) Functional and technological aspects of whey powder and whey protein products

Nastaj et al. (2020). Effect of erythritol on physicochemical properties of reformulated high protein meringues obtained from whey protein isolate.

use full names and abbreviations in the summary when you first mention them – PPI, SMP

Methods

How were the protein  solutions prepared?, basing on the pure protein calculation?

Materials – please mention all of the materials you use, you have forgotten to provide info on GDL, ethanol and so on

Line 61- was a gift from..

Line 62 – was a gift from…

do not use such repetitions – please write instead:  materials were kindly delivered by courtesy of…

I have doubts if sodium azide is needed? of course, if you have to dissolve the mixture overnight at room temperature…. Can’t you just do it at refrigeration temperature and  in this way limit the growth of bacteria? besides, if you want to use it in food systems, I think this additive will negatively affect the yogurt making process

give some space to all the mathematical equations and their descriptions. as they stand, they are not very readable. 

In the methodology for rheological tests, you mention that you are analysing G' and G'', and when describing the results you mention G*. I think it would be useful to present the course of these studies in graphs, not just selected values in tables   

Line 144 – “whey drained after” – this is not whey

Line 181 – it was not found in other studies

Line 191 – one period is enough

Line 193 - Figure 2 shows/pictures/presents…. In line you use abbreviation Fig 2. Please adjust abbreviations to MDPI standards

Line 238-241- please reword the sentence

Line 244 – dimension – aspect

Line 278 – Dumpler and Kulozik [16] and Huppertz et al [32] –

use a similar model afterwards in the paper when combining two references together

Line  313  - [28]  - [28]

326 – might contribute / might have contributed

times in tables 2 and 3, first in seconds, then in minutes and this is in decimal scale, please correct

which conditions, which mixture resulted in the fastest gel formation?

Paragraph 3.2.2

Line 397-399 please cite an articles on the interaction of whey protein with casein in yogurt, the literature you suggest is slightly outdated

Jørgensen et al (2015) Improving structure and rheology of high protein, low fat yoghurt with undenaturated whey proteins. International Dairy Journal 47 6–18

Mahomud et al (2017) Physical, rheological and microstructural properties of whey protein enriched yoghurt influenced by the heating milk at different pH values. Journal of Food Processing and Preservation 6(41) 1–8.

Nastaj et al. (2019). Physicochemical properties of High-Protein-Set Yoghurts obtained with the addition of whey protein preparations

Conclusions – too general

Please articulate and highlight the most important observations,

what applications  in the food systems do you see for the tested systems?

Line 461 – emulsions e foams?

Good luck with the corrections

Author Response

RESPONSE TO REVIEWER 3 COMMENTS

The article penned by Oliveira et al. is interesting and important to the industry. its strong point is the discussion of the results. However, I have a linguistic comments. in this form the article cannot be accepted. The text has to undergo English editing by native speaker in terms of grammar and punctuation. The article in current form is not a reader friendly.  there are a lot of errors in the text, there is no point in listing them all, I will just to name a few. Errors if not corrected will detract from your interesting paper. There are many commas in the text that are used unjustifiably.  Also, please don't use this many semicolons, combine sentences with each other more neatly. I recommend major revision.

Some other comments to improve the manuscript:

  1. Line 29 - partial solution to the global demand

It was corrected in the new version of the manuscript (line 31).

  1. Line 36 -…as beany limit its application

It was corrected in the new version of the manuscript, i.e., the quotes were removed (line 38).

  1. Line 45 – there is a lack of current publications on special product fortification technologies with whey protein preparations and using their functional properties. please cite:

Shayanti Minj and Sanjeev Anand (2020) Whey Proteins and Its Derivatives: Bioactivity, Functionality, and Current Applications

Kristensen et al (2021) Improved food functional properties of pea protein isolate in blends and co-precipitates with whey protein isolate

Ghanimah (2018) Functional and technological aspects of whey powder and whey protein products

Nastaj et al. (2020). Effect of erythritol on physicochemical properties of reformulated high protein meringues obtained from whey protein isolate.

The works of Shayanti Minj and Sanjeev Anand (2020) and Ghanimah (2018) were inserted as references in line 47, as they are related to the functional properties of milk/whey proteins.

The work of Kristensen et al (2021) was inserted as a reference in line 49, as it is related to the functional properties of mixed protein systems (pea and whey proteins).

We decided not to include the work of Nastaj et al. (2020) because it has a different focus and raw material compared to our work.

  1. use full names and abbreviations in the summary when you first mention them – PPI, SMP

As the template file of Foods does not contain a summary section, we used full names and abbreviations in the first mention to them in the Materials and Methods section (lines 64 and 65). In this regard, we followed other papers published in Foods as an example.

Methods

  1. How were the protein solutions prepared? basing on the pure protein calculation?

The rehydration step was based on the protein percentages of the powders employed in our work, being 83% (w/w) for PPI and 35% (w/w) for SMP. We explained the preparation of the suspensions in section 2.2.1, from lines 70 to 79, as shown below. We think that the text is clear enough to allow the reproduction of the work. However, if Reviewer 3 understands that this part of the paper can be improved, we are ready to do it following his/her advices.

“The SMP was rehydrated at 3% w/w proteins in distillated water. Sodium azide was added at 0.03% w/w to prevent microbial growth, and the sample was stirred at room temperature for 1 h. To generate the mixed protein suspensions, the PPI was added to the rehydrated skimmed milk with 3% (w/w) proteins to reach total protein concentrations of 5, 7, 9, and 11% (w/w). The same procedure was applied to prepare pure skimmed milk samples with the same total protein concentrations. With this strategy, there were nine samples in total, and the skimmed milk with 3% (w/w) proteins was used as a control sample (Table 1). All the protein suspensions were stirred overnight at room temperature, and then stored at 4ºC prior analysis.”

  1. Materials – please mention all of the materials you use, you have forgotten to provide info on GDL, ethanol and so on

The Reviewer is right. It was corrected in the new version, as follows: ethanol (Labsynth – São Paulo, Brazil) – line 125; GDL (Sigma Aldrich - São Paulo, Brazil) – lines 137 and 138. We did not identify other chemicals, instruments and software lacking producer information.

  1. Line 61- was a gift from..

Line 62 – was a gift from…

do not use such repetitions – please write instead:  materials were kindly delivered by courtesy of…

It was corrected in the new version. In order to avoid repetitions, we kept the first “gift from” for PPI and we used “kindly delivered by courtesy of” for SMP (lines 64 and 65).

  1. I have doubts if sodium azide is needed? of course, if you have to dissolve the mixture overnight at room temperature…. Can’t you just do it at refrigeration temperature and  in this way limit the growth of bacteria? besides, if you want to use it in food systems, I think this additive will negatively affect the yogurt making process.

We understand the Reviewer’s viewpoint. However, it was a deliberate choice we make to avoid bacterial growth that could affect our results. Our decision was based on the fact that, at the concentration of sodium azide employed in our work, the physicochemical properties of the suspensions are not affected. At the same time, we think that the Reviewer’s viewpoint is a valuable advice for future works using lactic acid bacteria.

  1. give some space to all the mathematical equations and their descriptions. as they stand, they are not very readable. 

We inserted additional lines before, between and after equations (1) and (2) (lines 96 – 100).

  1. In the methodology for rheological tests, you mention that you are analysing G' and G'', and when describing the results you mention G*. I think it would be useful to present the course of these studies in graphs, not just selected values in tables.

As a matter of fact, we used only G* in our illustrations (Fig. 6 and 7) in order to have clearer graphics. Nevertheless, we also recognize that the presentation of G’ and G’’ is important for the readers. Thus, as G* values are already presented in Table 3 and Fig. 6, we replaced G* in Fig. 7 by G’ and G’’ (line 460).

  1. Line 144 – “whey drained after” – this is not whey

The expression “whey drained after” was replaced by “liquid drained after” (line 151).

  1. Line 181 – it was not found in other studies

The sentence was corrected in the new version (lines 188 and 189).

  1. Line 191 – one period is enough

Two commas were removed in order to improve the clarity of the text (lines 191 – 194).

  1. Line 193 - Figure 2 shows/pictures/presents…. In line you use abbreviation Fig 2. Please adjust abbreviations to MDPI standards

All abbreviations “Fig.” were replaced by “Figure”, respecting the MDPI standards.

  1. Line 238-241- please reword the sentence

The sentence was shortened to improve its clarity (lines 244 – 246).

  1. Line 244 – dimension– aspect

The word “dimension” was replaced by “aspect” in line 253.

  1. Line 278 – Dumpler and Kulozik [16] and Huppertz et al [32] –

It was corrected in line 290.

  1. use a similar model afterwards in the paper when combining two references together

Line  313  - [28]  - [28]

It was corrected in line 326.

  1. 326 – might contribute / might have contributed

It was corrected in line 338.

  1. times in tables 2 and 3, first in seconds, then in minutes and this is in decimal scale, please correct

The values of time were standardized in seconds (Table 3).

  1. which conditions, which mixture resulted in the fastest gel formation?

The gel point values are presented in Table 3, and the discussion of the observed results is given from lines 365 to 385. Concerning the pure milk samples, the increase in protein content delayed the gel point (tg), and thus the fastest gel formation was observed for the control sample at 3% (w/w) protein. However, the addition of pea proteins induced a faster gel formation. For the mixed samples, the fastest gel formation was observed at 11% (w/w) protein, but we were not able to determine tg for this sample because when the rheometer started to measure the rheological properties, it was already gelled. This incident was explained from lines 381 to 385.

Paragraph 3.2.2

  1. Line 397-399 please cite an article on the interaction of whey protein with casein in yogurt, the literature you suggest is slightly outdated

Jørgensen et al (2015) Improving structure and rheology of high protein, low fat yoghurt with undenaturated whey proteins. International Dairy Journal 47 6–18

Mahomud et al (2017) Physical, rheological and microstructural properties of whey protein enriched yoghurt influenced by the heating milk at different pH values. Journal of Food Processing and Preservation 6(41) 1–8.

Nastaj et al. (2019). Physicochemical properties of High-Protein-Set Yoghurts obtained with the addition of whey protein preparations

We understand the Reviewer’s viewpoint, and considering the heat treatment, pH and protein concentration employed by Mahomud et al (2017), we decided to include this reference in our work (lines 413 and 416).

  1. Conclusions – too general

Please articulate and highlight the most important observations,

what applications  in the food systems do you see for the tested systems?

We included information about the main results of our work from lines 472 to 475, and also possible future applications for mixed protein systems, taking into account the limitations of the insoluble fraction of PPI for the development of new food products (lines 481 – 485).

  1. Line 461 – emulsions e foams?

It was corrected in the new version (line 481).

Round 2

Reviewer 2 Report

The authors have addressed the comments and revised the manuscripts accordingly.

Author Response

Thank you.

Reviewer 3 Report

Dear Authors,

The authors have implemented most of the substantive revisions I suggested. Thank you.

Author Response

Thank you.